# On the Road to Net Zero Health Care Systems: Governance for Sustainable Health Care in the United Kingdom and Germany

**DOI:** 10.3390/ijerph191912167

**Published:** 2022-09-26

**Authors:** Léa Weimann, Edda Weimann

**Affiliations:** 1Edinburgh Law School, University of Edinburgh, Edinburgh EH8 9YL, UK; 2School of Geography and Sustainable Development, University of St Andrews, St Andrews KY16 9AJ, UK; 3School of International Relations, University of St Andrews, St Andrews KY16 9AJ, UK; 4School of IT, University of Cape Town, Cape Town 7700, South Africa; 5Children’s Hospital Schwabing, Technical University Munich, 80804 Munich, Germany

**Keywords:** governance, healthcare, systems-thinking, net zero, sustainable development, public health, NHS UK, German healthcare, healthcare law, planetary health

## Abstract

Health care lies at the forefront of the impacts of climate change. Since the health sector is a major polluting and emission intensive sector, it remains a crucial challenge to address sustainability. The English National Health System (NHS) aims to be the first in the world to achieve net zero in all emission classes (Scope 1–3). In Germany, sustainability in health care is being driven bottom-up, while the Federal Ministry of Health at the time of the research in early 2021 takes no active stance on a net zero health care system. This article analyses the approaches to sustainability in the two different health care systems, explores common challenges, and draws recommendations to support the transition of the sector to a net zero future. An exploratory mixed method approach was taken applying qualitative and quantitative methods. This includes high-level expert interviews and an online survey from the United Kingdom (UK) and Germany. Results reveal the complex nature of health care systems and the need for engraining a systems-thinking approach. The findings call for the legal embedding of sustainability into the key principles of health care in Germany, endorses the ambition of the national health care systems in the UK, recommends collaborative cross-sector approaches for sustainability, and highlights the need for increased public awareness on the interrelation between human and planetary health to enable governance for sustainable health care.

## 1. Introduction

Planetary health is conceptualised as the interlink between human health and its dependence on nature and healthy ecosystems [1]. The current global planetary crises, such as COVID-19, environmental destruction, and climate change, require global action and collaboration across all sectors and scales [2]. The COVID-19 pandemic has shown how vulnerable society can be if hit by a global pandemic. Public health specialists call for improved sustainability and environmental protection to prevent future pandemics [3,4]. The Lancet Commission already warned in 2009 that “*climate change is the biggest global health threat of the 21st century*” [5] (p. 1659). Climate change is the biggest risk to health care through changing patterns of disease, causing water and food insecurity, extreme weather events (e.g., heat waves), forced population migration, and economic collapse [5,6]. As the foundation of our society, resilient and ecologically sustainable health care systems must be equipped for future challenges.

However, currently, the health care sector is pollution and emission intensive [7]. Yet, despite its high carbon footprint, it is relatively new to the concepts and principles of sustainable development [1,7]. Sustainable development has been most prominently defined in the Brundtland Report as “development that meets the needs of the present without compromising the ability of future generations to meet their own needs” [8]. While medical professionals are bound to the Hippocratic oath “first, do no harm” [7], it is only in recent years that a wider awareness has emerged within health care how this relates to sustainability [4]. Recent data calculations reveal that “*if the health sector were a country, it would be the fifth-largest emitter on the planet*” [7] (p. 4). Health care systems are rooted in unsustainable structures that are highly energy-demanding and socially impactful [9]. This is further complicated by an ever-growing population, finite supply of resources and raw materials, a growing share of elderly people, and lifestyle diseases [9]. Nonetheless, the health care sector has a responsibility to align its actions and developments to the United Nations (UN) Paris Climate Agreement [7]. However, there are difficulties around communicating and implementing sustainable development due to its complexity [10]. In the past, health care services have paid little attention to sustainability, since they are often overstrained in providing front-line services and frequently have historical notions of doing things [10]. This has led to organisational conservatism, vested interest, the feeling of health professionals that sustainability is not part of their job description, and a moral offset since they are already such essential contributors to health improvement [10]. However, health care services and their professionals undeniably must carry a major burden of unsustainable practices and lifestyles leading to an increase in non-communicable diseases and, therefore, becoming frontline workers of the climate crisis [7]. Consequently, there is a call for overarching sustainability structures that support sustainable development in the health care sector [11].

There is presently a limited amount of research on governance for a sustainable health care sector [9,11]. However, the first estimate of health care’s global climate footprint in 2019 revealed that the health care sector contributes to 4.4% of global net emissions and highlights the importance of governments in establishing action plans to decarbonise their health care systems, nurture climate change resilience, and improve health outcomes [7]. Sustainable development has also been taken up by the World Health Organisation (WHO), whose reports and resolutions stress the importance of addressing climate change [12]. Thus, sustainable development of health care systems is an emerging and interdisciplinary field that needs further attention and research.

This research study analyses and compares the sustainability governance approaches of the United Kingdom’s (UK) National Healthcare System (NHS) and the health care sector in Germany. In addition, it aims to explore common challenges to sustainability in health care and draw conclusions to support the transition of the health care sector to a net zero future. Both Germany and the UK are classified as major health care emitters in Europe, with a health care carbon footprint of 0.71t per capita and 0.66t per capita, respectively [7]. However, both countries have very different health care systems (Bismarck versus Beveridge Model) that approach sustainable development in various ways. This research explores key challenges that these health care systems face in becoming sustainable. The main research question for this study is:


*“How are the UK and Germany equipped to support their health care sector in becoming more sustainable and what are the underlying governance structures and challenges?”*


The Voluntary National Reviews (VNR) of European countries unveiled that there is limited discussion around the co-benefits on health through wider action on sustainability [13]. Themes such as governance, leadership, and multi-partner cooperation were frequently addressed, but health-related benefits were not included in the wider sustainability framing [13]. This indicates a governance and research gap for addressing and interlinking health concerns with environmental and sustainability issues.

Health systems are incredibly complex and there are no simple models of health systems and politics [14,15]. When comparing health care systems internationally, it becomes evident that there are various approaches to their set-up, organisation, optimisation, and efficiency. The various health care models include the Bismarck, the Beveridge, the National Health Insurance, the Private Health Care Insurance, and the Out-of-Pocket Payment Model [16,17]. The NHS and the German Health Care System can be used as representative models of the Beveridge and the Bismarck health care system. Good governance for health requires governments to make difficult and often structurally challenging choices [14]. This can only be achieved through transparency, accountability, participation, and policy capacity, as some of the key building blocks for good governance [14]. However, there are structural challenges to develop informed, useful, and evidence-based policy [15]. Governance problems include difficulties in long-term planning, badly thought-through policies, nepotism, incompetence, lack of trust, regulatory capture, corruption, and misaligned incentives [14]. Furthermore, there is an increased financial pressure on many European health systems that challenges the sector to deliver better health services but with fewer resources [14]. This poses a potential challenge with regards to implementing the structural changes needed for sustainable health care systems. Morris [15] emphasises the necessity of governance arrangements that can oversee and coordinate action. This is also mirrored in the WHO’s call for the implementation of whole-society and whole-government approaches to health and well-being [12].

However, improving governance in health care is a complex and enduring challenge [18]. The main governance models that are considered in this research are bottom-up and top-down governance, as well as a combined approach. Top-down governance approaches tend to focus on regulatory frameworks, while bottom-up approaches look at capacity-building and persuasive encouragement [18]. The bottom-up approach converges on grassroots initiatives and solutions through networks of activists, communities, and organisations [19]. Top-down approaches are important for mandating changes and setting regulatory frameworks, but bottom-up approaches are often seen as crucial for sustainable development since they engage local stakeholders and adjust regulations to local circumstances and create a local buy-in [20]. Thus, often the success of sustainability initiatives depends on long-term support and drive for those working on the ground and who are directly impacted by regulations but also environmental stresses [20]. Approaches to sustainability increasingly aim to balance and combine top-down and bottom-up governance [21].

The governance gap for sustainable health care has partly been addressed through the formation and emergence of Non-Governmental Organisations (NGOs), such as Health Care Without Harm (HCWH), which intend to implement sustainability at the heart of health care and mobilise health care’s political, economic, and societal influence to create a more sustainable, equitable, and healthy world [22]. This is realised through the projects of Global Green and Healthy Hospitals (GGHH), Safer Pharma, and Clean Med, which are implemented in countries across the world and provide a framework, action plan, and resources for health care systems and health professionals to implement sustainability [22]. Nevertheless, there is still a challenge around national governance to enable sustainable and ecological responsible health care, especially since environmental sustainability is not traditionally included within the principles of health care.

Health care in the UK (Beveridge Model) is publicly funded through a tax-based system, while in Germany (Bismarck Model), the government stays out of the day-to-day business of health care [16,23]. Hence, responsibility is delegated to institutions themselves and funding occurs through the health care insurance schemes [16,23]. This leads to a very different governance structure within the two health care systems [16,23]. In the Beveridge model, health care is universally provided to all and is financed by the government through tax payments [16,23]. As a result, government expenditure on health care is much higher in the UK than it is in Germany [16,23]. This is significant, since it plays into the governance of health care systems, and therefore, also into the implementation process of sustainability.

This crucially different governance structure of a national health care system has enabled the UK to establish structures for sustainability, such as the NHS England Sustainable Development Unit, which was created in 2008 [10,24] and has now been replaced by the Greener NHS Unit [25]. While the sustainability governance structure is slightly different in the various NHSs throughout the UK, the health system structure is nonetheless similar, thus allowing for a more involved governance approach to health care and sustainability in the health sector. In comparison, there is no overarching strategy for sustainable health care in Germany, since the government remains separate from the day-to-day business of hospitals. Hospitals have a much stronger institutional autonomy with no overarching obligation or strategy for sustainable development [11].

Health care models are an important consideration when analysing the governance for sustainability aspects within health care systems. In Germany, there is a very regional and decentred approach to health care with a key principle of self-governance [26]. It is an open public health care system that is set up centrally based on a statutory health insurance as well as a private health insurance system [27]. Health insurance in Germany is compulsory, and overall, the health care system is responsible for the health of around 83 million citizens [27]. The statutory health insurance carries around 87% of the population in Germany and makes up the biggest money pot [26]. It is divided into many insurance carriers, and these are bundled together in associations which are overseen by one overarching federal association [26]. These associations work together with the service providers to make contracts and agreements around patient services and resources [26]. The agreements are all made under the statutory system, which means that there is a public legal obligation that the service providers must fulfil [26]. This system lends itself to a strong centralisation of power and the power is the hands of those with the money. However, this system has a set of basic principles which guarantee that the ensured is provided with satisfactory health care based on sufficiency, functionality, and economics [26]. Sustainability is not yet included as a basic principle within this system. In addition, there is also the private health insurance system, which is a much easier built-up system and operates like a reimbursement scheme. Privately insured citizens make contracts that lay out the services which they are provided with [27]. There is a basis in the private health insurance system which determines the key services that must be ensured and received [26]. Due to its self-governing nature, there is limited state control over the health system in Germany, which makes governing quite complicated [26]. Caused by its strong regional and centralised approach, this poses a challenge for embedding sustainability nationally within the governance system of health care.

In comparison, the tax-based NHS is directly funded and mandated by the government, resulting in the state being much more involved in the governance aspect of health care [16]. Nonetheless, while the focus is often on the English NHS, it also important to be aware of the UK’s regional approach to health care, since there are four NHS systems in the UK: NHS England, NHS Scotland, NHS Northern Ireland, and NHS Wales, which are each controlled by the regional government. Whilst built up similarly on the Beveridge model, there are some regional differences to how far and to what extend sustainability is addressed in each NHS. In 2020, NHS England announced the goal to become the first national health system in the world to achieve net zero emissions, including all three emission scopes [28].

## 2. Methodology and Methods

### 2.1. Theoretical Research Framing

This research project leans towards a post-positivist epistemology and rejects the purely positivist research paradigm, which subscribes to the belief that knowledge is objective and can be abstracted from observation and research [29]. Rather, it acknowledges that research is imperfect and socio-political factors, values, and bias play into the social creation of knowledge [29]. From a constructivist understanding, the researcher co-creates knowledge, and thus, unconsciously includes their own perceptions and perspectives in the research process [30]. Nevertheless, for the purpose of this research, a pragmatic approach was taken. The pragmatic paradigm believes that the external world reality is both independent of the mind as well as rooted within the mind [31]. The pragmatic approach places a strong emphasis on value-orientation and practical theory [31]. It rejects reductionism of thought, culture, and psychological influences, while still recognising the importance of figures, statistics, and numbers [31]. Pragmatism lends itself to a mixed method approach and draws both from quantitative and qualitative research methods [29,31]. The combining of research approaches [32] tends to lead to a more complete and whole-rounded picture and can help break down complex real-world situations [31]. This is particularly relevant for sustainable development since it deals with such complex, multi-layered, and interdisciplinary issues.

The mixed method approach comprises a sequential exploratory design [30], which means that more emphasis is given to the qualitative aspect of research with quantitative data to backup and enrich the picture [29]. Overall, the research embraces a social constructivist epistemology, which argues that social meanings are varied, multiple, and depend on the subjective lived experience of individuals [31]. It draws on the rich qualitative responses from expert interviews while using the quantitative data from the survey to back up and enrich these answers, while also increasing the sample size of expert respondents on this topic and giving a wider range of perspectives including those working on the ground to implement sustainability in health care. The research is designed in a cross-case comparison of two cases according to case study research [33]. The governance approaches to sustainability in health care in the UK and Germany were compared and analysed.

### 2.2. Mixed Method Research Approach

The qualitative data are drawn from 11 expert interviews (Appendix A), of which five experts from Germany and six from the UK were interviewed; four were females and seven were males. The personal data of these experts are pseudonymised, generalised, and coded to protect the identity of the interviewees. Nonetheless, it is the high-level expertise that was shared for this research which makes it credible and useful for envisioning the future of sustainable healthcare governance.

Sampling was an important consideration for the research, since it determines the validity of the research and the statements that can be made about it [29]. Interviewees were recruited through convenience [34] and snowball sampling [29]. Through this sampling expert interviewees were identified that were available for interviews and these experts also helped point out colleagues who could contribute to the research. The interview followed a semi-structured approach using an interview guide to ensure that relevant topics were covered but also to allow for additional unplanned questions according to the specific expertise of the interviewee [29]. All interviews were conducted remotely via video call and in the preferred language of the interviewee to allow for full expression of interviewees (English or German). As a result, the interviewees that were experts in Germany were all conducted in German and then transcribed and translated by the researcher. In addition, a pilot interview was conducted in Germany to adjust and test the questions. Interviews were structured to be around 30 min, but many experts gave detailed answers and continued the discussion for up to an hour. Since the NHS is devolved in the UK, attention was paid to sampling expert interviewees from the different NHSs (two from Scotland, one from Northern Ireland, and three from England). There was no interviewee from Wales available; instead, secondary data were drawn from the sustainability strategy published in March 2021 by NHS Wales [35]. Nonetheless, since the structure of the different NHSs are similar, conclusions that were drawn about governance for sustainability will prove useful for all NHSs in the UK.

To back up the qualitative findings, secondary data, especially from the UK’s zero emission aim of the NHS, were analysed. Furthermore, an online survey (see Appendix B), available both in English and German, with both quantitative and qualitative questions, was sent out via various mailing lists in Germany and the UK to professionals engaged in the topic of sustainability in health care. The survey was started by 126 people, but only 84 (66.6%) of these responses were fully completed. Thus, the data include 55 responses from Germany and 29 responses from the UK. The research survey was conducted in March and April 2021, a time during which both health care systems were under considerable strain due to the COVID-19 pandemic. The survey provides a snapshot of health care professionals that play a pivotal role in shaping the future sustainability landscape in both healthcare systems. The aim was to enrich and support the answers of the expert interviewees with additional quantitative data from experts and staff working on this topic in their respective country. The survey was mailed out via the Health Care Without Harm (HCWH) network, the KLUG Alliance in Germany, and was also shared by health care professionals in their work environments. Since the survey was sent out to professionals and experts looking and working on sustainability in health care, it is important to be aware that there is a strong ecological awareness and bias towards sustainability in the responses. This is further shown by the fact that 96% of respondents rated addressing sustainability as important to extremely important. The focus of sending survey to those that are already interested in this topic area was intentional. The results by no means aim to claim or represent the opinions of either health care system. Rather, it aims to enrich the data and answers given by expert interviewees and those actively involved in addressing and implementing sustainability in health care. This proved particularly important in Germany, since there is no national governance system for sustainable health care, and thus, there are fewer secondary data available on this topic than there are in the UK. Hence, the high response rate in Germany was useful and indicated the interest in this topic area of governance for sustainable health care. In total, 65% of the participants were from Germany, while 35% were from the UK.

### 2.3. Data Analysis

For the qualitative data, thematic analysis was used for the coding of the interviews [36]. For example, it was coded according to the themes of importance given to sustainability, challenges mentioned, impact of COVID-19 on sustainability, and governance structures and approaches to sustainability. Specific attention was given in the rigorous transcription and translation of the interviews which form the foundation of the results, since the interviews were conducted in two languages and in two different health care systems. In the quantitative data analysis, descriptive statistics was used, looking at the distribution of data and focussing on summary statistics and visual representation [37]. For example, data analysis is conducted comparing the averages of importance given to sustainability in various system levels in the two countries. Overall, the aim of the research is to undercover the structures and challenges in both systems and draw recommendations for the vision of sustainable, ecological, and resilient health care systems.

## 3. Results

The analysed data are drawn from the high-level expert interviews, the survey, as well as secondary data sources and national commitments and policies. We will first compare the perceived importance and recognition of sustainability as a key issue in the health care sector in the UK and Germany. This is done through analysing the national commitments made to sustainability as well as looking at the survey responses and expert interview answers in both countries. Then, the key challenges for sustainable health care governance that were highlighted by interviewees and in the survey are explored and analysed. This gives an overview of the complex nature of health care systems and the many challenges it faces which call for a system-thinking response. The next section addresses the impact of the COVID-19 pandemic and analyses what respondents and interviewees have to say on how this impacted sustainability measures in health care. Furthermore, the structures and progress for sustainability are explored in the German health care system, with specific focus on what experts suggest for creating a governance structure for sustainability in the health care system. This is also explored in the English, Scottish, Welsh, and Northern Irish NHS. An overall picture of governance for sustainability in the NHS is given as well as a differentiation between the various regional approaches. The governance approaches in Germany and the UK are then analysed and compared by looking at top-down, bottom-up, and combined approaches of governance for sustainability. The aim of this results section is to answer the research question how the UK and Germany are equipped to support the health care sector in becoming more sustainable and what the underlying structures and challenges are.

### 3.1. Importance of Sustainability in the Health Care Sector: UK versus Germany

In October 2020, NHS England released its vision for a “net zero” health care service, aiming to be the first National Health Service in the world to achieve this target [28]. In the foreword the NHS Chief Executive Sir Simon Stevens announces:


*“The burden of the coronavirus has been exacerbated and amplified by wider, deep-seated social, economic and health concerns. The right response is therefore not to duck or defer action on the long-term challenges even as we continue to respond to immediate pressures. It is to confront them head on.*



*One of the most important is the climate emergency, which is also a health emergency. Unabated it will disrupt care and affect patients and the public at every stage of our lives. With poor environmental health contributing to major diseases including cardiac problems, asthma and cancer, our efforts must be accelerated.*


*We therefore make no apologies for pushing for progress in this area while continuing to confront coronavirus”* [28].

This highlights the importance of addressing sustainability in the health care sector and the need for transformational action. However, while NHS England has taken a pioneering and ambitious stance to address climate change, awareness for the importance of sustainable health care is not necessarily ubiquitous yet, and especially not on executive governance levels of other countries. Germany is an example of this, since the German Federal Ministry of Health and the German health care sector have not acknowledged or announced sustainability as a key topic or priority. This was also pointed out and confirmed by all five German experts.

As the quotes in Box 1 illustrate, Germany clearly seems to be lagging behind not only with regards to health care commitments to sustainability but also its recognition as a key topic at the national level. In Germany, the topic is growing in importance and recognition, but the UK has a clear vantage point by having a national mandate for addressing sustainability in health care.

Box 1Expert interviewees from Germany about the health ministry’s recognition of sustainability.DE.1 *“In comparison to the NHS, we in Germany are surely 5–6 years behind with regards to sustainability.”*DE.2 *“There is nothing really happening on this in the health ministry.”*DE.3 *“Currently we don’t have a plan like that in Germany but there are approaches in the pipeline.”*DE.4 *“Obviously, we here are not supported by the government. Thus, we realised we have to start NGO work and create change and a movement on the civil society level to help politics also move into this direction.”*DE.5 *“That is a problem in Germany since the Federal Ministry of Health has not yet understood that climate protection is a topic.”*

Furthermore, data from the survey (Figure 1) highlight this by comparing the perceived importance and recognition of sustainability as a key issue in the health care sector in the UK and Germany in a three-level format: system, hospital, and personal levels. The data illustrate that personal importance of sustainability is similar in both countries. UK participants personally ranked the importance of sustainability in health care on average at 91% and German participants at 88%. This figure then plummets to around 56% for hospitals in the UK and around 27% to hospitals in Germany. Looking at the health care sector, the UK respondents rank sustainability in the health care sector at around 59% and the German respondents ranked it at around 27%. This highlights that for participants in both countries, the importance of sustainability in the sector and in hospitals vs personal importance does not yet align.

Based on the survey, only 16% of German participants have sustainability in their job description and 13% partially. For UK participants, these figures sit at 79% and 17%, respectively (Figure 2). As already prefaced in the methodology section, this is obviously not a representative figure for either country. However, since the survey was sent out via health care and sustainability NGOs and spread through sustainability initiatives in health care, this should mean that there are similar types of participants and mainly people that are already interested and engaged in this topic area. Technically, this likely means a high range of participants with sustainability in their job description. However, this was not the case for German participants. Thus, the 16% of people that have sustainability in their job description in the German health care sector and the 79% in the UK are on the high end of the scale. The real figure for health care professionals in both countries is likely to be substantially lower. Especially in Germany, this highlights the low level of systemic inclusion of sustainability in health care.

### 3.2. Key Challenges for Embedding Sustainability into the Health Care Systems

All interviewees from both countries agree that the health care sector faces many challenges. The answers are illustrated in word clouds. The more the word is mentioned, the bigger it is. The word clouds in Figure 3 indicate the most repeated challenges that were mentioned by both interviewees and from the open-ended survey question. The word clouds illustrate economic considerations, with regards to cost, the financing system of health care, and resources in general, which are the key considerations in both countries. Thus, the economic challenge of sustainability and net zero is similarly evident in both countries.

Awareness is still also identified as a key challenge. As expert UK.4 puts it in the interview:


*“The last major challenge that I would identify is the public perception that health care should just be about frontline health care work and should not have anything to do with sustainability. Anything that we do that is not seen to be directly related to frontline health care we will get it in the neck for. If we did a report on, for example, fixing a boiler and greening the estate, you will have some people going ‘you have just spent seven million quid on a new boiler when really my granny died, and you could have put the money in to save her’.”*


This quote indicates a key challenge for changing perceptions around sustainability and the importance of addressing it in the health care sector.

These challenges are not unique to either country and overall, the challenges in the health care sector seem to be similar regardless of health care structure. Especially key concerns around economics and costs persist in both health care systems, even though they are financed differently as outlined above.

Even if one were to take climate change out of the picture, the health care sector already faces many challenges [38], which COVID-19 has further highlighted and worsened [4]. Thus, a systems-thinking approach is needed to address them because changes in one area will have consequences in another [38]. As experts highlight in Box 2, the challenge of truly integrating sustainability into the health care system to achieve net zero is difficult.

Box 2UK expert quotes about the challenge of integrating sustainability across the health care system.UK.1 *“Having done this for 9 years, having been working with health care providers to try and make this happen, seeing the challenges, seeing how difficult it is… yeah, I think it is a great challenge, but I think it is ambitious.”*UK.2 *“Systems thinking is very important. You need to have horizon scanning and adjustments—it is a living process.”*UK.5 *“How to deliver an integrated approach to sustainability in a system with many silos is a big challenge.”*

### 3.3. The Impact of COVID-19 on Sustainability in Health Care

According to survey participants and expert interviewees COVID-19 has had a significant impact on the health care sector, and therefore, on sustainability in health care. This research was conducted amidst the COVID-19 pandemic, with both countries facing rising COVID-19 infections and deaths. In the survey COVID-19 was mentioned as a key challenge for sustainability in Germany three times and in the UK four times. Expert interviewees further spoke about the impact of COVID on the health care sector and both the positive and negative narrative this has placed on sustainability. This is further illustrated through the survey responses (Figure 4).

Figure 4 shows that German participants rated COVID to affect sustainability, with 8% of participants stating that sustainability is not on the agenda anymore, 48% stating it was never on the agenda, and 8% saying that COVID-19 has significantly impacted sustainability progress.

In the UK, this was slightly different, with 2% stating sustainability is not on the agenda anymore, 11% saying it was never on the agenda, and 19% saying that it has significantly impacted sustainability progress (Figure 4).

In addition to the more negative impacts, 15% of participants in Germany rated sustainability to have encouraged professionals to explore new sustainability initiatives, and for the UK, this figure even rises to 27%. While respondents differed in their opinion on the effect COVID has had on sustainability, only 7% in Germany and 3% in the UK stated that it had little to no impact. Thus, COVID-19 has undoubtedly contributed to the narrative and future of the health care sector with relation to sustainability.

Furthermore, there is the stark contrast between 48% of respondents in Germany expressing that sustainability has never been a priority on the agenda, and only 11% of UK respondents confirming the same. This highlights a major governance difference with regards to the importance that is placed on sustainability within the health care sector in both countries.

The COVID-19 pandemic has put a massive strain on the health care sector. Public health care specialists as well as epidemiologists warn that the future likely holds more pandemics that might be more serious and deadly [3]. The two main things that were mentioned by interviewees in the UK are the impact and increase of waste and single-use plastic waste due to COVID-19 infection control, but also the positive impact of things such as increased use of online consultation and less business travel for those that are not frontline workers. Thus, even in the health care sector, there have been some positive developments in terms of recognising the interlink between planetary and human health [4] as well as pushing for innovation, change, and new ways of doing things. Interesting to note is what interviewee DE.4 from Germany explains with regards to the cultural developments of society caused by fear:


*“We can learn from COVID-19 that if there is a certain element of fear at play then we will change. However, personally I am of the opinion, that it is not a good cultural development if change is only triggered through fear. Ultimately fear does not progress us as society except through causing a lot of suffering. It might progress us in some technical and institutional aspects, but not in the humanistic way. We need to develop new spiritual awakenings in people.”*


Thus, while COVID-19 should serve as a lesson for society the changes initiated by the pandemic, they are by no certainty a healthy and long-term sustainable development for society.

### 3.4. Governance for Sustainability in the German Health Care System

The previous sections have already illustrated that there are currently no notable national governance structures for sustainability in the German health care system. Interview questions explored whether a national plan for sustainability exists, what structures there are for sustainability, and whether sustainability was being addressed through bottom-up or top-down governance (see Appendix A). Experts in Germany agree that sustainability is being driven from a bottom-up grassroots approach (Box 3).

Box 3Experts about the governance approach to sustainability in Germany.DE.1 *“It is a bottom-up governance approach to sustainability. There are individual hospitals that do outstanding work and progress with regards to sustainability, but these are driven by leadership forces within the hospital and so called ‘sustainability champions’.”*DE.2 *“Mostly, it starts through individual people who are interested in the topic and who will link up internally with people by asking who else is interested in the topic.”*DE.3 *“Politically there are a few incentives that encourage sustainability in health care but in comparison there are only few. (…) However, overall, when looking at the whole system these are only small initiatives and incentives.”*DE.4 *“Definitely, bottom-up governance.”*DE.5 *“I would say it is bottom-up. (…) There are scattered sustainability initiatives of hospitals aiming to become sustainable. (…) However, all these are individual initiatives and we definitely need a legal foundation for sustainability in health care.”*

The survey data also illustrate that most participants agree that sustainability is approached through bottom-up governance (Figure 5). In Germany, top-down governance is often understood as governance from the hospital executive board or hospital management. Hence, there is a differentiated understanding of what is meant by top-down governance between the two countries, since the health care system in Germany is not managed and regulated through a national system, as in the UK. Thus, top-down governance in relation to health care is often understood with regards to executive buy-in.

Nonetheless, top-down governance is clearly important to enable widespread action and incentivising commitments to sustainability. There is also a strong recognition that bottom-up involvement is crucial for the success of implementing sustainability in health care. This was also pointed out by experts in the interviews that recognise bottom-up governance as essential for implementing sustainability into hospitals. Furthermore, there is consensus that Germany needs a legal and regulatory framework as well as incentives that supports hospitals in becoming more sustainable. There is a call to support the work that is being done by individual hospitals and initiatives. Currently, these are inhibited by the fact that economic considerations are prioritised, and hospitals are steered by the purpose of being resource efficient and profit-driven rather than focussed on holistic patient-treatment and environmental sustainability. The quotes in Box 4 point out that hospitals across the country need the financial and legal mandate to make decisions based on sustainability rather than on cost.

Box 4Experts about the economic focus of the health care system in Germany.DE.1 *“We need legislation, enforcement, and financial motivation that supports and obligates hospitals to address sustainability such as certifications or health insurance committing to sustainability.”*DE.3 *“The German health care system is very centred around the economic costs. Economic considerations definitely trump ecological ones.”; “These hospital groups and companies are on the stock market, so any money spends needs to potentially be justified to shareholders.”*DE.4 *“Our health care system in Germany is really good but our population is in no ways healthier. Why is that?—Money and investment. Money makes the world go round.”*DE.5 *“Money also plays a role in arguing for sustainability. Even though sustainability can often save money there is still a pre-conception that is costs money and capital investment.”; “Many sustainability benefits are not material in that sense and that can be a problem since we live in a deeply capitalist system.”*

Thus, to address this challenge of not having a legal or regulatory basis that supports and promotes sustainability in the German health care system, one health care law specialist (DE.3) that was interviewed presented the following options for change. Firstly, there is the option of embedding sustainability in the key principles of health care. Secondly, one could try to embed sustainability at every single point and service of the health care system, (e.g., ambulance, pharmacy, health service resource providers, etc). Thirdly, there is the option of embedding a new law into the constitution. The following paragraphs present the challenges and feasibility of these legal options, as explained by DE.3.

Embedding sustainability into the key principles of health care would mean that the highest courts, such as the Federal Social Court, would take on the sustainability principle and every health service measured accordingly. This would create the functional details for the health care system through the judicial system. It would then allow the service providers to utilise money in the health care system based on sustainability. The difficulty lies in predicting the form this would take, since the Federal Social Court is not an easy system and is also politically loaded, which can easily become complicated and politically charged. Nonetheless, this could practically work in legally supporting sustainability actions and responses within health care.

The option of embedding sustainability at every single point and service of the health care system would mean embedding specific sustainability principles based on services and CO_2_ emissions. However, for this to work, this would entail micro-managing every single point in the health care system. This has both advantages and disadvantages. The advantage is that it allows for very specific actions and goals right from the start. The disadvantage is that it is difficult to predict the consequences and effects of the embedded sustainability goals. If it is set in law but then it turns out that it is not functional, it would mean that it would need to be reversed. However, reversing a law is even more complicating then first putting it into place. It would have to go through the whole political system, and this would take considerable time. It would then slow sustainability action, which is time sensitive, as it is and raise doubts about its feasibility within the health care sector.

Thirdly, the most contested legal option is writing a new law into the constitution. Currently, in the German Constitution, Article 20(a) lays out the life conditions for every citizen, but that is an empty clause in the law. It is a so-called argumentative clause that technically does not enforce obligations. Arguing for sustainability action because of this clause specifically with reference to the health care system is likely to be cumbersome and unsuccessful. To change these specific conditions about sustainability and climate protection, they would have to be included in the constitution. Furthermore, once such a new clause is included in the constitution, it is likely to be permanent, and therefore, there is great political caution in constitutional changes. Additionally, there is the challenge of needing a majority political consensus. Thus, in general, lawyers tend to avoid attempting to change the constitution.

All three of the discussed legal options to support sustainability in the German health care system at a systemic level present complex challenges. However, the health care law specialists (DE.3) recommends that option 1 would be the most feasible stating:


*“I think the first option of embedding sustainability into the key principles of health care law is the most realistic and best starting point right now. It would make sense to try embedding this not only into health insurance law, but into the law of all social insurance.”*



*“This would be the most realistic but also not without challenges since the legislators would realise the intensions behind the change and the fact that this would give new power to the social courts and law. This would give social courts an argumentative pattern which they could and would use politically. Social law is political.”*



*“All of this has pros and cons, but we need to create a change because we cannot carry on watching how it’s going wrong right now.”*


Germany is clearly lagging behind on enacting and enabling governance for sustainability in comparison to the UK. Financial constraints make it more difficult for those trying to enact sustainability in the current health care system. Besides, the complex structure of the German health care system makes mandating sustainability into health care across the country challenging, especially as it is not a national system such as the NHS. Nonetheless, that does not mean that sustainability cannot be embedded through other mechanisms such as incentives, certificates, sustainability audits, and importantly through health care law. This is a challenge that must be addressed taking into consideration a system thinking approach that makes sure sustainability is addressed holistically through the lens of clinical health and patient and staff wellbeing.

### 3.5. Governance for Sustainable Health Care in the UK

NHS England is pioneering sustainability in health and has the ambition to become the first net zero national health care system in the world. The NHS “Delivering a ‘Net Zero’ National Health Service” report illustrates that a recent survey shows that 98% of all NHS staff believe that more action on sustainability in health care is needed [28]. This indicates a broad consensus among the medical profession for action on climate change in England. Furthermore, there is also a recognition that the upskilling of staff will be needed to embed sustainability into everyday actions in health care services [28]. With the new ambitious commitment to net zero health care, NHS England is clearly ahead of the game, but as expert UK.2 points out (Box 5), there is a risk that the NHS becomes complacent because it now has a plan and is pioneering sustainability in health care. Even within NHS England, there are still a lot of challenges ahead as well as uncertainties around how to realise this ambition.

Box 5Challenges and risks for the implementation of the net zero strategy in the UK.UK.2 *“I think the biggest risk or danger for the NHS now is that they get complacent and don’t do as much because ‘they have got a plan’. This is a journey and you have always got to be able to adjust and adapt and be one or two steps ahead of the game or wave to prepare that moment.”*UK.5 *“There is a lot of discussion around greater strategy at the level of the Greener NHS but when it comes to on-the ground implementation there is less information on what that means for on the ground so that is where individual trusts and knowledge networks on the topic come into play.”*

In Figure 5, the survey data reveal that 62% of participants agree that the approach to sustainability in the UK is a combined approach and that it is in line with what the expert interviewees from England agree on. They explain that sustainability is addressed through a top-down approach with regards to strategy and a more bottom-up approach with regards to implementation (Box 6).

Box 6Expert views about top-down and bottom-up governance approaches in the UK.UK.1 *“It is a mix of both bottom-up and top-down.”*UK.2 *“It is definitely a mix, and it needs to be a mix. You need the bottom-up because you need people on the ground doing things because that is when it becomes real but sometimes people need that extra to convince the board.”*UK.5 *“From a national perspective it is balanced. The top-down approach that they are taking in terms of the targets is good, but the practical side tends to be tackled bottom-up.”*

Nonetheless, expert UK.5 highlights that while bottom-up approaches are essential, more coordination would be desirable, due to the nuanced nature of sustainability and difficulties around determining which options are truly more sustainable:


*“There might be several different answers to one question on sustainability. For example, which product is the most sustainable? You might spend three days of research to discover that from the carbon perspective it is this product, from an environmental perspective another, from a social value perspective it is this, from a modern slavery perspective it is yet another product–it is a nuanced topic and that makes it all much harder. To be fair on Greener NHS it is a challenge to lay out the exact recommendations for what is best and most sustainable.”*


Thus, governance for sustainability in the English NHS are taken through a combined bottom-up and top-down approach. This is not without challenges, and as interviewee UK.3 pinpoints, “*Not to say that the NHS is a system that is perfect and that is what is driving sustainability. However, it does help that we have a national health care system.*” Nonetheless, even within the English NHS, sustainability is not systemic yet, and the challenge of a system-thinking approach that engrains sustainability throughout all health care services remains.

While NHS Scotland is similar in structure to NHS England, their sustainability actions and goals differ. NHS Scotland has at the time point of this study not yet published a net zero sustainability strategy, but it is set to be published soon. However, in 2019, when the Scottish government declared a Climate Emergency, NHS Scotland made six high-level commitments which the chief executives endorsed [39].

Overall, the Scottish NHS targets align with the government net zero commitment by 2045. The exception to this is at that point in time are the vague commitment towards Scope 3 supply chain emissions. Expert UK.4 points out that the public sector is obliged to meet Scottish government targets. Furthermore, sustainability action in the public sector is coordinated through the Sustainable Scotland Network, a space for sharing practice and steering sustainability action. The governance approach started out primarily as top-down but is now a combination of top-down and bottom-up governance for sustainability with increasing body of on-the-ground staff driving sustainability (Table 1). Nonetheless, a major challenge identified is both related to the financing of net zero as well as unavoidable emissions, such as anaesthetic gases:


*“With defining net zero, no definition has been provided by the Scottish government, so it is not known what is included. We include waste, water, anaesthetic gases, and f-gases which are Scope 1 and 2. Scope 3 includes water, waste, and business travel. Staff commuting, patient travel and supply chain are excluded which is controversial as these emissions dwarf others, but they cannot be measured accurately. England and Wales did it, but it is deeply flawed. They assign kilograms of carbon per counts spent. So, if 10% of the cost is saved, then 10% of carbon is reduced—this is rubbish.”*



*“Nonetheless, we are including the supply chain in our sustainability strategy and there is a huge amount of work happening with suppliers and with staff commuting (…) but essentially all of those are things that are beyond our control. Thus, it would be difficult to include that within a numeric net zero target. Again, we are not excluding it from our strategy, but we are not including it within our ’numbers’.”*



*“Changes for sustainability in the health care sector are not supported financially by the government. This is not unique to the health sector. Scottish government published their latest Climate Change plan update in December 2020 and there are big ambitions in there when they are going to decarbonise heat, fleet, etc. but the financial support and packages behind it are not good enough.”*


NHS Northern Ireland does not have a net zero strategy for their health care sector. However, they are nonetheless bound to the government’s commitment to net zero. In comparison to the NHS England and NHS Scotland approach to sustainability, the current focus seems to be on bottom-up governance. However, it is likely that the new Climate Change Bill in Northern Ireland will mandate a new and stronger approach to sustainability in the NHS.

This research does not include an expert interview with a representative from NHS Wales, caused by non-availability, but uses secondary data that the NHS Wales published, i.e., the “Decarbonisation Strategic Delivery Plan”. The strategy was set up together with the Carbon Trust which works with business and governments to align their climate strategies to meet the goals of the Paris Agreement [35]. The strategy currently states the aim to be net zero by 2030. This was mandated through the Climate Emergency declaration by the Welsh government in 2019, and since the NHS in Wales makes up the largest public sector organisation, they have a large role in realising this goal [35]. They aim to address all three emission classes while openly acknowledging the unavoidable emission nature of some of their services, since their priority remains delivering safe and high-quality care [35]. Nonetheless, the NHS Wales Chief Executive shows a clear commitment and recognition of the risk Climate Change and issues such as air pollution put on public health. This illustrates executive buy-in, and thus, likely the support of top-down governance approaches to sustainability. To what extend this is supported and complemented by bottom-up approaches cannot be deducted from the literature. Nonetheless, the strategic plan strongly suggests a combined approach, since over 100 industry experts and health care professions were consulted and contributed to inform the strategic delivery plan, which includes 46 initiatives and targets for decarbonisation [35].

Something that is to be commended from the NHS Wales strategy is that it includes short-term targets for 2020–2022, 2022–2026, and 2026–2030 [35]. This breaks down the task and allows for adjustments and regular evaluation. Furthermore, a re-evaluation and assessment are planned in for 2025 and 2030 [35]. In addition, there is a strong government legislative and ambition-orientated approach to sustainability in Wales, as can be seen in Table 2. This means that sustainability action is strongly supported and mandated across the public sector in Wales, which is important for a systems approach to sustainability that addresses all scopes of emissions and includes everyone. This is also highlighted within the net zero strategy documents, which state that the most critical step is that all parts of the NHS Wales engage with sustainability, collaborate, and support the ambitions on the road to net zero [35].

## 4. Discussion

Overall, there is a consensus about an organised approach for governance for sustainability in the health care system in the UK. The opposite is true for Germany (see quotes in Table 3). The following table presents quotes about organised governance approaches for sustainability in health care comparing the UK and Germany.

Figure 6 illustrates that 61% of UK survey respondents either agree or strongly agree that there is an organised governance approach. This picture is flipped in Germany where 69% either disagree or strongly disagree with the statement that there is an organised governance approach to sustainability (Figure 6). Furthermore, Figure 6 shows 0% of respondents in Germany strongly agree that there is an organised governance approach for sustainability in health care and 0% of respondent in the UK strongly disagree with the statement that there is an organised governance approach. This illustrates that Germany and the UK paint two completely different pictures of how sustainability in health care is addressed. For the health care system in Germany this should be a strong signal and call for action. Moreover, it confirms that the UK is ahead of the game. Nevertheless, experts and survey respondents highlight that there are still major challenges ahead and these are not too dissimilar in the two health care systems.

Experts in the UK generally agree that there is an organised approach to governance for sustainability in health care, but also point out that it has evolved over time and the results are still mixed. Sustainability is not engrained across the system into all trusts and hospitals yet. This can be seen, for example, through the climate manager initiatives, which aims to have a staff member in trusts and hospitals across the country that is responsible for helping to establish sustainability and support the transition to net zero. While this has become widespread in the NHS, it is not systemic yet (Box 7).

Box 7Sustainability managers in the NHS.UK.2 (England) *“There are some organisations that employed sustainability managers on the understanding that they would save more than their salary. That was quite a pressure for sustainability managers. It’s quite a pressure to come into a role and job and know you need to save the hospital more money that you cost. However, generally that worked and what is happening now is that these people are gaining seniority and it is being recognised. That is why you don’t just have a junior person who is only working in the engineering room, but you want someone who can actually work at a more senior level and can have a relationship with the board or even beyond the board.”*UK.4 (Scotland) *“Each NHS board in 2012 was meant to appoint two people—a sustainability champion at an executive board level and also somebody at an operational level but the effectiveness and successful of these roles differ. Some of the sustainability champions at executive board level are very proactive and some of them get it and then some other boards aren’t even aware of the sustainability board champion.”*UK.5 (England) *“I would not say that all trusts have sustainability managers. It varies and what they mean by sustainability is also very different. You might have an energy manager whose job it is to run an energy saving campaign or you might have a waste manager who is in charge of recycling but that is different and depends on your definition of sustainability.”*UK.2 *“It is not yet systematic in UK everywhere but what you might see is that there is an energy or waste manager who is also asked to look at sustainability.”*

The survey explored this further by asking participants if they feel that there is a support system for them to realise sustainability initiatives (Figure 7). Figure 7 reveals that the majority agree that there is some sort of support system, with 17% strongly agreeing and 55% somewhat agreeing. Nonetheless, this is also not systemic yet, with 14% somewhat disagreeing, 4% strongly disagreeing, and 10% neither agreeing nor disagreeing. This would be an interesting point of research to explore further to identify how this could be improved.

The German participants of the survey (Figure 7) reveal a puzzling result for the same question, almost exactly splitting the response on if they feel that there is a support system, with 7% both strongly agreeing and strongly disagreeing, 27% somewhat agreeing and 24% somewhat disagreeing, and 35% neither agreeing nor disagreeing. This polar result opens an interesting question to explore in future research. A suggested explanation could be that while there is no obvious support system, there is a growing network of bottom-up approaches to sustainability in health care which allow for sharing, organising, and supporting individuals in their sustainability endeavours across the country. These include networks such as HCWH [22], KLUG [40], and KLIK Green [41]. Furthermore, NGOs such as KLUG have recently set up new collaborative initiative together with other organisations to strive for net zero through a bottom-up approach to which hospitals can sign up [42].

Similar to the NHS in the UK, there is also an initiative in Germany (KLIK Green) that supports the training of climate managers in hospitals. This initiative is one of the few sustainability initiatives in health care that is national in scope and that is also supported by the German Federal Ministry of Environment. Yet, it is not widespread and there is no mandate for hospitals to have a climate manager or to address sustainability at all. As interviewee DE.5 explains in Box 8, the KLIK Green initiative currently supports just over 250 hospitals in training sustainability managers, whereas there are around 1900 hospitals in Germany in total [27]. As explained by expert interviewee DE.5 in the quotes below (Box 8), this is an initiative that could be rolled out nationally, but political pressure would need to be build up to mandate it.

Box 8Quotations regarding the climate manager initiative in German hospitals.DE.5 *“KLIK Green is a qualification project to train climate managers in hospitals.”*DE.5 *“The KLIK project was very successful and that is why the federal ministry for environment came to us and asked if we couldn’t do this for 500 hospitals but we said we can’t manage that simply just the mission of mobilising 500 hospitals is a huge task so we set for the goal of 250 which was already incredibly ambitious. Now we actually managed to mobilise more than 250 hospitals and that despite COVID-19.”*DE.3 *“There are very few hospitals in Germany that employ sustainability managers. Furthermore, this is a challenge because hospitals need to economically justify such positions.”*DE.5 *“I could imagine that it could be implemented both regionally across the states in Germany as well as at the national level. However, for that a lot of political pressure would need to be created. I could see it happening, but I think it will take at least another 3–5 years.”*

Overall, we conclude that the road to net zero and creating organised governance approaches for sustainable health care is still rocky and uncertain. Nonetheless, strides have been taken in both countries to try and address this. Germany’s response is still mainly grassroots-led, and more needs to happen to support this through regulation and legislation at the national level to cut carbon emissions to reach net zero. The UK has clearly taken ambitious steps for mandating net zero from the executive level. However, experts also point out that this is still in the early stages and not systemic yet. Box 9 concludes this results section, with some final quotes from experts in Germany and the UK pointing out the urgency, the need for system thinking, and the challenge of fundamentally transforming the health care sector.

Box 9Concluding thoughts from expert interviewees in Germany and the UK.DE.1 *“No, it is not about bringing it onto the agenda in the long-run—we have to address it now!”*DE.5 *“I think there are a lot of approaches and possible solutions, but we have to look at the whole system overall to find whole-rounded solutions.”*UK.2 *“It is not just about driving down the carbon footprint but achieving the right of healthy people on a healthy planet.”*UK.2 *“The challenge is that we will have to fundamentally transform the health sector if we are going to do this well…”*

## 5. Conclusions

### 5.1. Key Findings

Governance for sustainable health care in Germany is led through a bottom-up governance approach with individual climate champions leading initiatives. While bottom-up approaches are important for engraining sustainability into the system and creating changes on the ground, urgent action is needed from a political as well as governmental level to support such developments. The German Federal Ministry of Health and health insurance companies must support, encourage, and incentivise sustainability measures. Furthermore, creating a legal mandate for sustainability is necessary to address the economic constraints on current sustainability initiatives. In addition, hospitals and health care professionals must be guided, skilled, and supported to establish sustainability into the everyday systemic structure and routine of health care. It can be recommended that a way to do this would be through a health care law reform by including sustainability in the key principles of health care. This would prioritise sustainability over cost, and furthermore, would encourage a societal shift towards sustainable and ethical procurement. Furthermore, climate manager initiatives such as the KLIK Green project should be supported, expanded, and rolled out across the country.

Although there are regional differences to how sustainability is addressed in the UK’s various NHSs, governance for sustainable health care is chiefly led through a combination of bottom-up and top-down governance. This combined top-down, and bottom-up approach is desirable, since it mandates changes from the executive level, and through this, gives support to sustainability initiatives. In addition, bottom-up action also means that there is on the ground support and staff that are leading these changes. The challenge lies with making this ambition truly systemic, involving all those working in health care and being prepared to create large transformational changes in the way health care currently functions.

Key common challenges for sustainable health care in both countries were identified as economics and costs, additional funding, as well as awareness around how health care and sustainability interlink. The financial constraints need to be seriously considered and raised at a political and government level. Education about climate change and its impact on health must improve to create widespread awareness about its importance to enable the societal changes that need to occur. Doctors and especially nurses are some of the most trusted professions in society and they can, therefore, help to lead awareness campaigns around climate change and the health impact of it [2,22]. More government support is needed with regards to financing sustainability and creating a sustainable infrastructure that allows for the creation of a net zero future. In addition, health care professionals must play a role in enhancing awareness about the health implications of climate change.

COVID-19 has impacted sustainability initiatives in both countries in various ways, both through deprioritising it as well as inspiring action and system transformation. COVID-19 has highlighted the vulnerability of society to infectious diseases and worsening planetary health [4]. The urgency for acting on climate change should not be underestimated, especially since the health care sector lies at the forefront of dealing with climate change. Thus, COVID-19 should be used a narrative for action and transformation to avoid future pandemics.

The ambition for net zero health care requires a systems-thinking approach that considers the big picture and looks for whole-rounded solutions by taking everyone along for the journey and engraining sustainability at the core of decision making. While net zero is an ambitious goal for a sector that is inevitably tied to emissions and is essential for society, nonetheless, human health can only be achieved through planetary health. Health care system need to re-evaluate whether they are just treating symptoms or truly healing. The transformation that is needed is no small ask and should be supported from every level of society.

### 5.2. Limitations and Future Research

Governance for sustainability in health care is a complex topic worthy of exploring further in longer and more detailed contexts, especially when looking at more than one health care system and model. It is recommended for future research to extend the cross-case comparison to other countries and health care models in Europe to further draw on different approaches to sustainability in health care. Similarly, the opposite could also be useful by diving more specifically into one country and health care system and expanding upon the challenges and possible solutions on how to address sustainability in that specific health care sector. Nonetheless, comparisons are always good for drawing inspiration and encouraging action and collaboration across the health care sector.

Both a strength and limitation of this research is that it took a convenience sampling approach that focussed specifically on consulting experts and survey participants that are interested and active in this topic area. This allowed for a rich analysis of sustainability in health care and recommendations how to improve it. Nonetheless, the limitation is that it these opinions and importance placed on sustainability are not representative of the whole sector. It would be interesting to explore the level of awareness around sustainability in health care further. In addition, a further research opportunity would be to analyse how demographics and especially age played into views, perceptions, and conceptualisations of sustainability within the health care sectors. Thus, it is recommended for future research to expand this study to include more voices on this topic and a larger proportion of the population. Furthermore, additional expert interviewees in Wales and Northern Ireland could enrich the UK regional picture.

In addition, it would be of great interest to conduct this research again post-COVID-19 to explore if the pandemic had a transformational effect on the health care sector and to analyse progress towards sustainability and net zero aspirations.

### 5.3. Final Remarks

To conclude with a quote by Mahatma Gandhi: “*It is health that is the real wealth, and not pieces of gold and silver*.” The economic focus on cost and profit-maximisation even in health care distract from the real wealth of a society, which cannot be measured through GDP but through the health and happiness of a population. The stresses on the health care sector can be relieved through the creation of a healthier society. By reducing the number of patients and procedures in hospitals, these measures directly reduce emissions and help to create a more resilient society and health care system. Health care lies at the forefront of the planetary crises and its resilience is the gate keeper to a stable future. Thus, it must address the governance challenge of a net zero transition. The normative power of health care systems and professionals should not be underestimated, and health care systems need to rise to the greatest challenge and call for transformation that society has ever faced. Crucially, this must also be supported by regulations and law. Humans are not separate from nature and the interconnection between human and planetary health must form the basis of transformation not just for the health care sector, but for all of society to ensure a future on a habitable planet.

## Figures and Tables

**Figure 1 ijerph-19-12167-f001:**
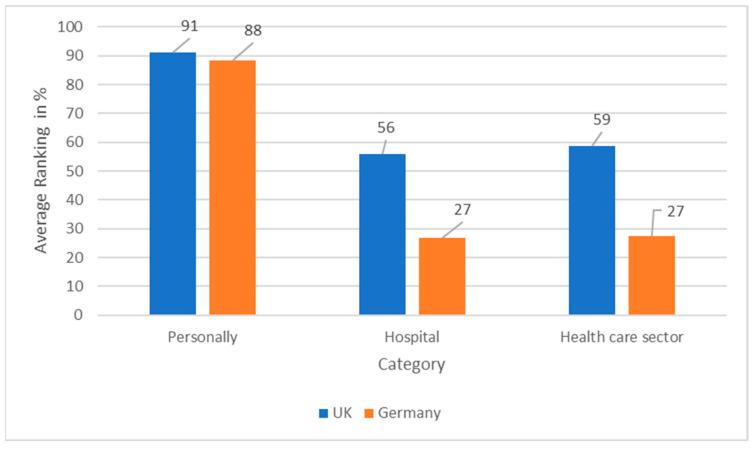
Bar graph illustrating the ranking of importance for sustainability from personal to health sector level.

**Figure 2 ijerph-19-12167-f002:**
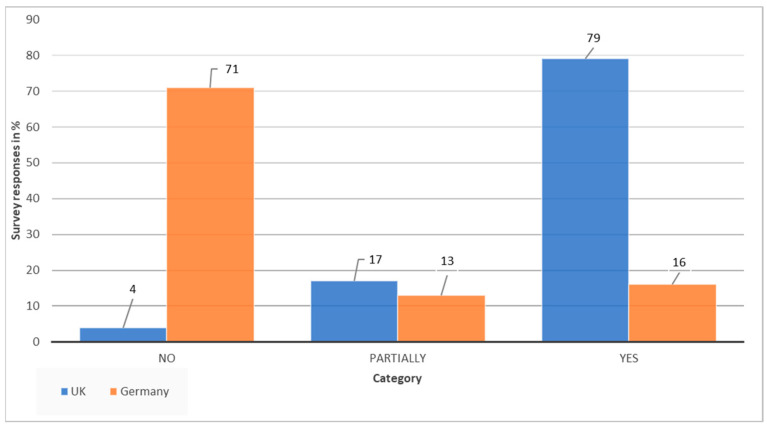
Comparison of UK and German participants with sustainability in their job description.

**Figure 3 ijerph-19-12167-f003:**
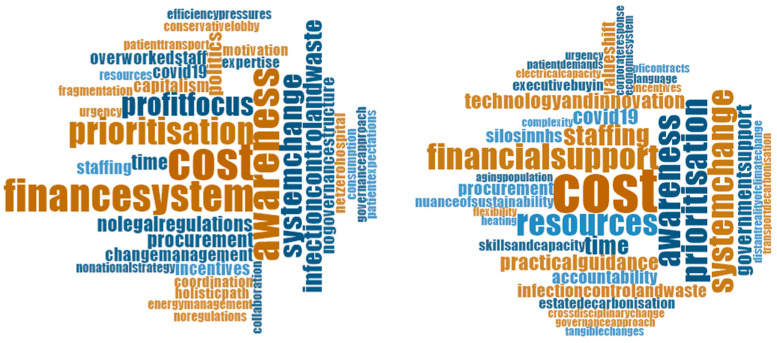
Word clouds of key challenges in Germany and the UK.

**Figure 4 ijerph-19-12167-f004:**
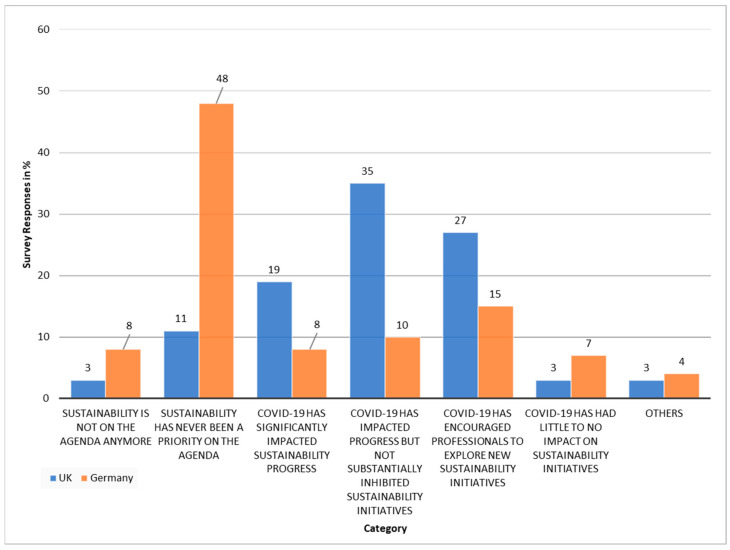
Comparison of UK and Germanys survey responses about the impact of COVID-19 on sustainability in health care.

**Figure 5 ijerph-19-12167-f005:**
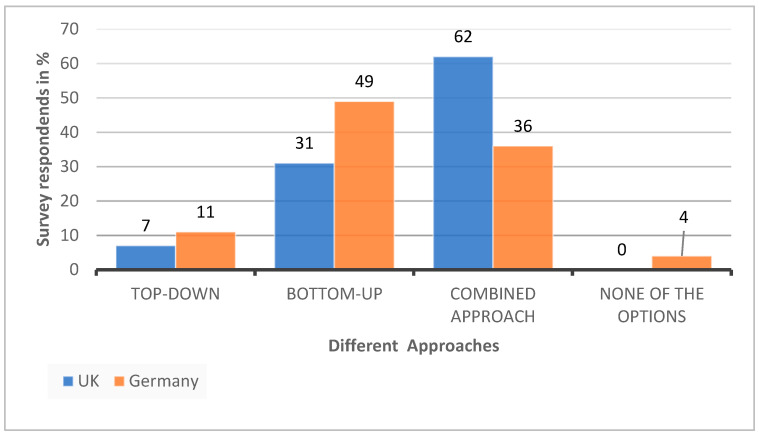
Comparison of UK’s and Germany’s survey results regarding the perceived governance approach.

**Figure 6 ijerph-19-12167-f006:**
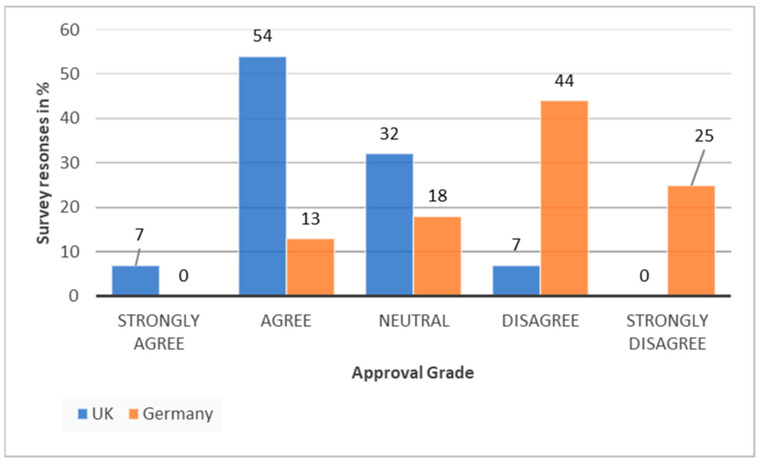
Comparison of the UK’s and Germany’s survey responses about the governance approach for sustainability.

**Figure 7 ijerph-19-12167-f007:**
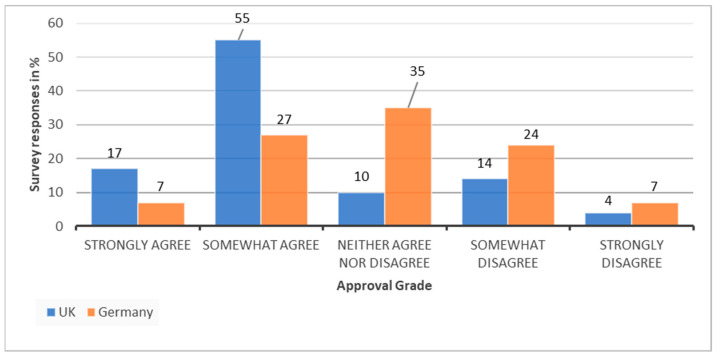
Comparison of the UK and German survey participants on support system for sustainability initiatives.

**Table 1 ijerph-19-12167-t001:** The six high-level commitments of NHS Scotland.

Recommendations by NHS Scotland Chief Executives (June 2019)
NHS Scotland is to be net zero with respect to greenhouse gas emissions by 2045.
2. All new NHS Scotland buildings and refurbishments are to be designed net zero from April 2020.
3. All NHS boards should perform a climate change risk assessment including all operational areas and developing a climate change adaptation plan aiming to develop resilience under changing climate conditions.
4. The NHS’s own transport fleet will be net zero by 2032.
5. Scope 3 emissions from supply chain will be analysed and reviewed for environmental impacts.
6. Each NHS board is to establish a climate change/sustainability governance group for the net zero transition.

**Table 2 ijerph-19-12167-t002:** Wales Decarbonisation Strategic Delivery Plan illustrating the Welsh government legislative and strategic structure for climate action [35].

Legislation (Wales)	Strategy (Wales)	Ministerial Ambition (Wales)
The 2015 Well-being of Future Generations Act	The 2017 “Prosperity for All—Economic Action Plan”	Net zero public sector by 2030
The 2016 Environment Act	The 2019 “Prosperity for All—A Low Carbon Wales”	70% of Wales electricity consumption is to be renewable by 2030
The 2018 Climate Change Regulations (Carbon Budgets)	The 2020 “Prosperity for All—A Climate Conscious Wales”	1 GW of electricity generated in Wales is to be locally owned by 2030
		By 2020 all new developments are to have an element of local ownership

**Table 3 ijerph-19-12167-t003:** Governance approach to sustainability UK vs. Germany.

UK	* 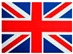 *	Germany	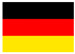
*UK.1 “The governance approach to sustainability in the NHS is helped by the way the NHS is organised and run as well as the culture of the NHS.”*	*DE.1 “There is no monitoring or steering process that governs for sustainability in health care.”*
*UK.3 “It is still in its early days, but it has been inspirational and we are hoping that it will inspire even more to step forward and follow suit based on that inspiration of net zero.”*	*DE.1 “If we manage to make the upcoming election a climate election and I think we are on a good path to do this- then regardless of who will end up being elected, we can set this topic into the election programmes of the different parties. If we manage to put this topic into the party programmes, then we can also bring this into legislation.”*
*UK.5 “Yes, because there is a national target and requirement. Although, I would say that it does not go down to every single NHS trust. So nationally yes but within every single trust–less so.”*	*DE.3 “We could nationally mandate hospitals across the country to incorporate sustainability. However, if not enough financial support is given hospitals and doctors could retaliate by making use of the ‘basic law of freedom of occupation’ since that would impact on their cost and incomes. Thus, these law changes would need to incorporate information and resources on how to finance this switch to sustainability.”*
*UK.4 (Scotland) “Yes there is but the adherence to it and support of it is very mixed. Some boards are great and other boards are dreadful.”*	*DE.1 “Through bringing it into legislation we can bring a governance structure for sustainability in health care.”*

## Data Availability

Not applicable.

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
