# Peer review of "On the Road to Net Zero Health Care Systems: Governance for Sustainable Health Care in the United Kingdom and Germany"

_ijerph, 2022, doi:10.3390/ijerph191912167_

Round 1
Reviewer 1 Report
Could the authors please explain why this part it si in line 133 and also in line 155, 170, 380?- Klicken oder tippen Sie hier, 133 um Text einzugeben..-
In line 337 the bibliography is not well inserted. Also in line 352, 353, 368, 402, 403, 410, 452, and also in ather lines were there is also another font.
Do the authors have permission to use the image presented as figure 6?
Please restructore more clean all the article because it is hard to read and understand with all the insertions.
It is not fluent and not very clear.
The article does not follow the structure of a research scientific article.
Author Response
Thank you for the review. We have worked on the language and flow. All sections were improved and made more clear. A special emphasis was put on the references. Figure 6 was amended. We aligned the flow in line with a scientific paper.
Reviewer 2 Report
Section 3.1 needs to be rewritten. I really did not get anything out of that section. It was very unclear to me
Basic demographic statistics would be useful in the results section
The word clouds in Figure 3 are not helpful. A table or graph would be much more helpful and useful to the reader
Many of the references are missing and show as Error Reference source not found. An easy fix.
I am not sure that Table 1 is very useful. I personally would remove it and prefer a written out explanation
Figure 8 is very difficult to read
Section VI - the table doesn't help, it's too informal. Write out the results and explain them in-depth
Author Response
Thank you for reviewing the manuscript.
As recommended Section 3.1 was rewritten.
Basic demographics were added when available. In the survey we did not ask for all demographics.
We deleted some boxes and figures to improve the flow and readability. We deceided to leave the word clouds as they highlights the amount of anwers regarding specific topics.
We improved the Bibliography so that now all references are included.
Table 1 was deleted and replaced as suggested by a written explanation.
We changed Figure 8 as well as the conclusion in section VI where we removed the table and wrote out the text in paragraphs instead.
Thank you for your comments and advice.
Reviewer 3 Report
The Authors raise an important issue of sustainable health care governance in United Kingdom and Germany. The subject matter is interesting and important also from the point of view of other countries.
My comments and suggestions for development:
Introduction
In the introduction, it would be useful to define what the Authors understand by the concept of sustainable development. It is also worth mentioning what the purpose of this work is, why the authors conduct such analyzes. The target appears only on lines 290-292.
- lines 130-133 - The Authors refer to the healthcare system in United Kingdom and Germany. In a literature review, it would be useful to refer to health care systems in other countries. Are these systems similar to each other? Are they financed in the same way?
- lines 133-134, 155, 170-171 - errors in editing
Methodology and Method
- line 226 - space between the article and (36)
- line 228 - double space
It is a pity that the authors of the study did not use the Cronbach's alpha reliability coefficient to assess the reliability of the questionnaire.
Results analysis and discussion
- lines 319, 334-336, 367-372, 388-389, 400-401, 432-433, 448, 476-477, 498-499, 516, 575 - various fonts
- lines 399, 412 - it is worth unifying COVID-19
- line 473 - table 1 requires editing correction
- figure 6 needs improvement
- figure 7 requires improvement, it is illegible
Comparative results discussion
- figure 8 - the drawing is illegible
- lines 612, 630, 642-645, 653-656 - various fonts
- line 616 - this is not a pie chart
Key findings, recommendations, limitations, and conclusion
- line 661 - various fonts
- table 2 does not meet the editing requirements
References - Not in accordance with the requirements of the Journal
Notes to drawings - drawings require standardization. The axes in the drawings should be described.
I hope that my insights will be useful for the Autors of the study.
Author Response
Thank you for reviewing the manuscript.
We used a mixed method approach for a qualitative-quantitative study. The described research methods are well established and appropriate for the topic.
As outlined in the manuscript health systems are very complex. We have chosen two representatives of major public health care systems: the Beveridge and the Bismarck system. The way how they are financed is complicated, especially the Bismarck system as private insurance companies are allowed as well. As the study already is quite detailed we did not elaborate on the Kutzin framework that explains fianancing mechanisms of healthcare systems to allow access for all. However, as recommended we included an introduction to different health care systems and explained sustainability and defined sustainable development according to the Brundtland Report.
The reason for the study is introduced in the abstract and in the introduction from line 42ff onwards (However, currently the healthcare sector is polluting and emission intensive....")
We adjusted the references, unified references to COVID-19, and improved the figures as well as reduced the number of figures. Thanks for all your advice.
Round 2
Reviewer 2 Report
The paper is much better.
In the file I received, the references were again not there - I believe this is a problem with the journal software as it does not link to whatever reference software you use. Completely not your problem or fault.
I would suggest that demographic information be collected in your surveys. To me this is a major limitation of your research. I won't say your paper should not be published, but I would strongly encourage you to get demographic information in the future.
Otherwise, I appreciate the corrections. Good luck in your future work!
Author Response
Thank you for your valuable input and comments.
We have included the demographic factor in the limitation section as a suggestion for future research. At the time we wanted to keep the survey short to enhance the response rate and there was no capacity to include additional analysis on demographics. We appreciate your comment and agree that demographics around age would add to the analysis.
In addition we have further edited the manuscript for minor improvements and typos.
Best wishes and thank you for reviewing the manuscript!